# OpenReview forum: "Not All LLM Reasoners Are Created Equal"
_NeurIPS.cc/2024/Workshop/MATH-AI — MATH-AI 24_

### Official Review · Reviewer_Aswf · 2024-10-06
**Review of "Not All LLM Reasoners Are Created Equal"**

**Rating:** 7
**Confidence:** 5

**Review:**

This paper presents a study on compositional GSM8K where two GSM8K examples are chained together. This paper presents a thorough study on one of the key questions for reasoning with LLMs, i.e., compositional generalization and multiple interesting observations. The appendix presents comprehensive experimental details and futher discussions such as thinking with natural language vs code.

---

### Official Review · Reviewer_xFy5 · 2024-10-07
**Intriguing case-study**

**Rating:** 7
**Confidence:** 3

**Review:**

**Summary**

This paper introduces a 2-hop reasoning version of GSM8k, *Compositional GSM*, consisting of pairs (Q1,Q2) of modified problems drawn from (a subset of) GSM8k where the final answer to Q1 is referred to as X which is used as a variable in Q2. They measure the "reasoning gap" as the difference in accuracy on (the subset of) GSM8k vs. the compositional version. The main result is that smaller models have a much greater reasoning gap than larger models.

**Comments**

References (\ref) from the main text to the relevant appendix sections should be added. I also think descriptions of figures should occur in the captions of the figures themselves, but perhaps this is just my opinion :)

What is instruction-tuning?

65: "We study four family of models"

What are the odds that the reasoning gap boils down to simple context-length? (I'm not an LLM expert -- perhaps smaller models simply don't have enough attention for a two-paragraph question?)

**Pros**

* Clearly written
* The reasoning gap has a simple and natural definition
* Shows a clear trend that model size is crucial for "reasoning capacity" even on tasks thought to be easy for small models
* Inspires further research into "composed questions" or some underlying concept of sequential reasoning
* I think the fact that larger models have a smaller reasoning gap is intriguing and deserves further investigation. Even if not, circumventing this gap with smallest models is surely of interest.

**Cons**
* Minimal investigation into why the reasoning gap occurs (trends in incorrect responses, etc.)
* "They often answer the first question correctly, but lose attention to details and make subtle errors in answering the second question." This should be justified by examples, if not a full experimental analysis.
* Details important for reproducibility are lacking (e.g., exactly how is the dataset constructed (all-pairs (Q1,Q2) from GSM8k)? Pretraining and instruction tuning details? Code?).
* Related work?
* Should the checklist not be attached to the appendices?

---

### Official Review · Reviewer_qWqv · 2024-10-07
**A compositional version of GSM8K that challenges LLMs**

**Rating:** 6
**Confidence:** 4

**Review:**

**Summary**

The paper presents a new version of the GSM8K benchmark, which is created by composing two questions together. The final answer for the first question is substituted as a variable into the second question, and the LLM is asked to solve the second question using the first answer substituted in for the variable.

**Strengths**
* The paper draws interesting insights about the difference between the performance and behavior of small vs. large models, base vs. instruction-tuned models, etc.
* The paper includes a detailed appendix with all prompting information, failure modes, and additional experiments that make it easy to conduct future work.
* The method proposed in the paper can be applied to any mathematical benchmark that relies on numerical calculations.
* The average accuracy of most models is considerably below what the expected average accuracy would be, showing an inherently more challenging task.

**Weaknesses**
* This is more of a suggestion, but it would be interesting to compare the performance of models on this benchmark to their performance on a 2-prompt setup of the benchmark (i.e. asking the first question, letting the model calculate the answer, and then asking the second question as “Let X be the answer to the previous question… [insert second question here]”)

---

### Decision · Program_Chairs · 2024-10-08

Accept